# Real-Time Generation of Time-Optimal Quadrotor Trajectories with Semi-Supervised Seq2Seq Learning

**Gilhyun Ryou, Ezra Tal, and Sertac Karaman**
Laboratory for Information and Decision Systems (LIDS),
Massachusetts Institute of Technology,
Cambridge, Massachusetts 02139
{ghryou, eatal, sertac}@mit.edu

**Abstract:** Generating time-optimal quadrotor trajectories is challenging due to the complex dynamics of high-speed, agile flight. In this paper, we propose a data-driven method for real-time time-optimal trajectory generation that is suitable for complicated system models. We utilize a temporal deep neural network with sequence-to-sequence learning to find the optimal trajectories for sequences of a variable number of waypoints. The model is efficiently trained in a semi-supervised manner by combining supervised pretraining using a minimum-snap baseline method with Bayesian optimization and reinforcement learning. Compared to the baseline method, the trained model generates up to 20 % faster trajectories at an order of magnitude less computational cost. The optimized trajectories are evaluated in simulation and real-world flight experiments, where the improvement is further demonstrated.

**Keywords:** Motion planning, Model learning

## 1 Introduction

Generating a feasible minimum-time quadrotor trajectory, i.e., the fastest trajectory that can be flown accurately and reliably on the actual vehicle, is challenging due to the complex nonlinear quadrotor dynamics. At high speeds, these dynamics include phenomena, such as unsteady aerodynamics, estimator error, and battery dynamics, that make it difficult to incorporate realistic feasibility bounds in trajectory generation. Popular methods address this difficulty by using simplified feasibility models [1, 2, 3] or by minimizing snap (i.e., the fourth temporal derivative of position) as a proxy for feasibility [4, 5]. Given a waypoint sequence and a time allocation over the segments between these waypoints, a corresponding polynomial minimum-snap trajectory can be obtained efficiently through convex optimization. However, finding the optimal, i.e., minimum-time and feasible, minimum-snap time allocation still amounts to a much harder nonlinear optimization problem.

Existing methods for finding (near-)optimal time allocations suffer from downsides with regard to applicability, evaluation time, or their ability to incorporate realistic feasibility constraints. For example, while gradient descent (with or without regularization) has been demonstrated with simplified dynamics [5, 6, 7], its large number of evaluations makes it computationally too expensive to consider a more realistic feasibility test. Additionally, the algorithm may not converge to a feasible solution altogether [8]. On the other hand, methods that are capable of incorporating complex realistic feasibility constraints are typically not suitable for online motion planning. For instance, their learned feasibility representation may be specific to a given sequence of waypoints [9].

In this paper, we address these shortcomings through a novel deep neural network model that generates the optimal minimum-snap time allocation for any given sequence of quadrotor waypoints. Specifically, we utilize a temporal neural network model with sequence-to-sequence (seq2seq) learning to handle sequences with a variable number of waypoints. Our work contains several contributions. Firstly, we design the deep neural network trajectory generation model and train it using a realistic six-degree-of-freedom flight dynamics feasibility model. The resulting model generates faster trajectories than standard minimum-snap methods at a fraction of the computational cost, making it suitable for high-rate online motion planning. Secondly, we propose a novel training process

6th Conference on Robot Learning (CoRL 2022), Auckland, New Zealand.

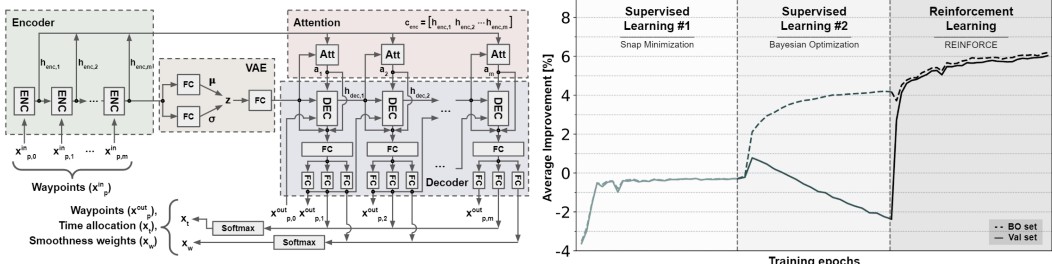

(a) The proposed seq2seq model

(b) Average improvement over training phases.

Figure 1: (a) The proposed model with fully-connected (FC) layers, bi-directional GRU encoder (ENC), and basic GRU decoder (DEC). Attention values are estimated from the context vector $\mathbf{c}_{\text{enc}}$, i.e., all encoder hidden states, and the preceding hidden state. The model takes the variable-length waypoint sequence $\mathbf{x}_p^{\text{in}}$ as input and outputs the time allocation $\mathbf{x}_t$ and the smoothness weights $\mathbf{x}_w$ as well as the reconstructed waypoint sequence $\mathbf{x}_p^{\text{out}}$. (b) The three-phase training for forward-facing trajectory with differential flatness feasibility constraint.

that utilizes Bayesian optimization to enable supervised pretraining on a data subset. This pretraining serves as a guiding step that increases the efficiency of reinforcement learning by improving the sampling quality. Thirdly, we extend the minimum-snap optimization formulation by adding an individual snap penalty weight for each trajectory segment, enabling more aggressive minimum-snap trajectories. Fourthly, we demonstrate the improvement of the proposed model over standard minimum-snap trajectory planning in both simulation and real-world flight experiments.

## 2 Preliminaries

### 2.1 Quadrotor Trajectory Planning

We study the problem of planning a time-optimal quadrotor trajectory that passes through $m + 1$ given waypoints $\tilde{\mathbf{p}} = \begin{bmatrix} \tilde{p}^0 & \tilde{p}^1 & \cdots & \tilde{p}^m \end{bmatrix}$. Each waypoint $\tilde{p}^i$ consists of a prescribed position $\tilde{p}_r^i$ and yaw angle $\tilde{p}_\psi^i$. A trajectory is a continuous function that maps time to position and yaw, i.e., $p(t) = \begin{bmatrix} p_r(t)^\mathsf{T} & p_\psi(t) \end{bmatrix}^\mathsf{T}$. The time-optimal planning problem can be formulated as follows:

$$\underset{p,\, T,\, \mathbf{x} \in \mathbb{R}_{\geq 0}^m}{\text{minimize}} \quad T \quad \text{subject to} \quad p(0) = \tilde{p}^0, \;\; p\left(\textstyle\sum_{j=1}^i x_j\right) = \tilde{p}^i, \; i = 1, \ldots, m,$$
$$T = \textstyle\sum_{i=1}^m x_i, \;\; p \in \mathcal{P}_T, \tag{1}$$

where $\mathbf{x} = \begin{bmatrix} x_1 & \cdots & x_m \end{bmatrix}$ is the time allocation over the $m$ segments between waypoints, and the function space $\mathcal{P}_T$ is the set of all feasible trajectories, i.e., all trajectory functions that the quadrotor can successfully track over $[0, T]$. This set is defined by physical phenomena (e.g., aerodynamics, thrust limitations, and vehicle mass and inertia), as well as hardware and software limitations of the estimation and control systems. Consequently, fast and agile motion planning is subject to a complex feasibility boundary that makes directly solving (1) challenging.

Widely used quadrotor motion planning algorithms employ minimum-snap trajectory generation [4, 5], which does not directly consider the complex feasibility bounds. Instead, it minimizes the fourth-order derivative of position, i.e., snap, and the yaw acceleration to generate smooth trajectories that are less likely to violate feasibility constraints, as follows:

$$\underset{p \in \mathbb{R}_{\geq 0}^m}{\text{minimize}} \quad \sigma\left(p, \textstyle\sum_{i=1}^m x_i\right) \quad \text{subject to} \quad p(0) = \tilde{p}^0, \;\; p\left(\textstyle\sum_{j=1}^i x_j\right) = \tilde{p}^i, \; i = 1, \ldots, m, \tag{2}$$

where

$$\sigma(p, T) = \int_0^T \mu_r \left\| \frac{d^4 p_r}{d^4 t} \right\|^2 + \mu_\psi \left( \frac{d^2 p_\psi}{d^2 t} \right)^2 dt \tag{3}$$

with $\mu_r$ and $\mu_\psi$ weighing parameters. By considering piecewise polynomial trajectories, the convex program (2) can be solved efficiently [5]. For convenience, we denote the resulting minimum-snap

trajectory $p = \chi(\mathbf{x}, \tilde{\mathbf{p}})$. One way to obtain the time allocation $\mathbf{x}$, which appears as a constraint in (2), is to first find the optimal allocation ratio as

$$\underset{\mathbf{x} \in \mathbb{R}^m_{\geq 0}}{\text{minimize}} \quad \sigma(\chi(\mathbf{x}, \tilde{\mathbf{p}}), T) \quad \text{subject to } T = \sum_{i=1}^{m} x_i. \tag{4}$$

Since the optimal allocation ratio is independent of the duration $T$ [5], a uniform scaling factor $k$ can be used to find the the fastest trajectory that satisfies the feasibility constraint, i.e., $\chi(k\mathbf{x}, \tilde{\mathbf{p}}) \in \mathcal{P}_{kT}$ [10]. This constraint can be evaluated in various ways, ranging from a simple check based on differential flatness of the idealized quadrotor dynamics [4] to experimental evaluation in flight tests [9]. While the resulting time allocation $k\mathbf{x}$ is on the feasibility boundary, it is typically not time-optimal because there may exist a different (non-minimum-snap) time allocation ratio that remains feasible for smaller $kT$.

## 2.2 Bayesian Optimization

Bayesian optimization (BO) is a class of machine learning algorithms to efficiently solve optimization problems with black-box objective or constraint functions that are expensive to evaluate. The Gaussian process classifier (GPC) is widely used as a *surrogate model* to approximate constraint functions based on (noisy) evaluations [11]. Each point is selected as the maximum of the *acquisition function* $\alpha(\mathbf{x}|\mathcal{D})$, which balances uncertainty reduction in the surrogate model against the anticipated improvement in the objective function, given all data $\mathcal{D}$ obtained in previous evaluations. Multi-fidelity Bayesian optimization (MFBO) extends the notion of BO to incorporate evaluations from various sources. The key idea is that combining cheap low-fidelity evaluation with expensive high-fidelity measurements improves overall efficiency. For instance, a rough simulation or an expert's opinion may serve as low-fidelity model, while a high-accuracy simulation or real-world experiment serves as high-fidelity model.

Ryou et al. [9] utilize MFBO to find the minimum-time allocation $\mathbf{x}$ for quadrotor trajectory generation. By efficiently combining evaluations from analytical models, numerical simulation, and real-world flight experiments, the algorithm finds faster trajectories that remain dynamically feasible on the actual vehicle, when compared to standard minimum-snap algorithms. However, the learned surrogate constraint model does not generalize beyond the predefined waypoint sequence $\tilde{\mathbf{p}}$, making the algorithm unsuitable for online motion planning.

## 2.3 Deep Learning-based Approaches

Deep learning-based approaches have been applied in the field of quadrotor motion planning and control, e.g., to improve model predictive control parameters through deep reinforcement learning [12, 13] and for neural network flight control [14, 15, 16, 17]. de Almeida et al. [8] apply deep learning to generate minimum-snap trajectories. They train a network model to generate the time allocation between the prescribed waypoints. By replacing the time-consuming nonlinear optimization (4) with neural network inference, an algorithm speed-up is achieved. However, since it uses a fully-connected network, the algorithm is limited to sequences with a specific number of waypoints. Additionally, the model is trained only with supervised learning using labels from minimum-snap optimization, so it does not provide minimum-time solutions.

Sequence-to-sequence models take a variable-length input sequence and transform it to an output sequence [18], which makes then particularly suitable for natural language processing applications [19, 20, 21, 22, 23, 24, 25]. Seq2seq learning has also been applied towards various motion planning problems, such as predicting the future trajectory from past states and control inputs [26, 27, 28, 29] and generating trajectories that are predictable for other agents, e.g., in multi-agent settings [30] and for socially-acceptable trajectory generation [31].

## 3 Algorithm

We propose seq2seq learning with a specialized training scheme to build a model that generates the time-optimal trajectory for any given sequence of waypoints. By using minimum-snap trajectories, we obtain a finite parameterization of the trajectory in terms of the time allocation over the segments between waypoints. Additionally, we introduce *smoothness weights* $\mathbf{x}_{wi}$, leading to the following

snap objective (cf. (3))

$$\sigma(p, \mathbf{x}_t, \mathbf{x}_w) = \sum_{i=1}^{m} x_{wi} \left( \int_{T_{i-1}}^{T_i} \mu_r \left\| \frac{d^4 p_r}{d^4 t} \right\|^2 + \mu_\psi \left( \frac{d^2 p_\psi}{d^2 t} \right)^2 dt \right) \tag{5}$$

where $T_i = \sum_{j=1}^{i} x_{tj}$, $\sum_{i=1}^{m} x_{ti} = m$, $\sum_{i=1}^{m} x_{wi} = m$ and $\mathbf{x}_{ti}$ refers to the time allocated to the segment between the $i$-th and $(i{+}1)$-th waypoints. The smoothness weights enable faster trajectories that remain feasible by allowing modification of the objective to locally increase aggressiveness, according to

$$\underset{\mathbf{x}_t, \mathbf{x}_w \in \mathbb{R}_{\geq 0}^{m}}{\text{minimize}} \; T, \quad \text{subject to} \; T = \sum_{i=1}^{m} x_{ti}, \; \chi(\mathbf{x}_t, \mathbf{x}_w, \tilde{\mathbf{p}}) \in \mathcal{P}_T. \tag{6}$$

Our goal is to build a model that outputs the optimal time allocation $\mathbf{x}_t$ and smoothness weights $\mathbf{x}_w$ for any given waypoint sequence $\tilde{\mathbf{p}}$ with variable length. Solely using supervised learning to train the model is impractical, as obtaining the optimal time allocation and smoothness weights with regard to a realistic feasibility model (i.e., solving (6)) is computationally expensive, e.g., taking almost an hour for a single set of waypoints using the BO approach from [9]. At the same time, reinforcement learning is inefficient and may fail, because randomly sampled time allocations and smoothness weights often fail to solve (5) altogether or require absurd paths with extreme excursions between waypoints. In order to address these challenges, we propose a three-phase training process consisting of pretraining, guiding, and generalizing. We first pretrain the model using supervised learning with minimum-snap labels obtained from (4), which is not time-optimal but gives realistic trajectories. Next, we guide the model towards better solutions using optimal time allocation and smoothness weights obtained using BO [9] for a subset of training dataset. The model exploits the progress from pretraining and guiding to efficiently sample in the final reinforcement learning phase, where the entire training dataset is used to generalize the policy. In the remainder of this section we describe the dataset, the seq2seq model, and the three training phases. Additional details are given in the appendix.

### 3.1  Waypoint Sequences Dataset Generation

We create a dataset of sensible waypoint sequences by randomly sampling sequences in the unit cube $[-0.5, 0.5]^3$ and accepting samples based on two topological constraints: the summed Menger curvature [32] (i.e., the reciprocal of the radius of the circle that passes through a subsequence of three waypoints) and the summed Euclidean distance between subsequent waypoints, as follows:

$$I_{\text{curvature}} = \sum_{i=0}^{m-2} \frac{1}{R(\tilde{p}_r^i, \tilde{p}_r^{i+1}, \tilde{p}_r^{i+2})} \in [5, 20], \quad I_{\text{distance}} = \sum_{i=0}^{m-1} d(\tilde{p}_r^i, \tilde{p}_r^{i+1}) \in [0, 30]. \tag{7}$$

An initial minimum-snap trajectory obtained from (4) is used to reject waypoint sequences that result in trajectories that go beyond $[-1, 1]^3$ and to set the yaw component for each waypoint tangential to the local velocity. We unwrap the reference yaw, so that the yaw difference between subsequent waypoints is less than $\pi$ rad. Finally, the actual waypoint positions are obtained by scaling with the desired space scale $L_{\text{space}}$.

### 3.2  Sequence-to-Sequence Model

The proposed seq2seq model, shown in Fig. 1a, consists of encoder, decoder, variational autoencoder (VAE), and attention modules. Encoder input is normalized as $x_{p,i}^{\text{in}} = [\tilde{p}_r^i/(L_{\text{space}}/2) \quad \cos(\tilde{p}_\psi^i) \quad \sin(\tilde{p}_\psi^i) \quad f_{\text{EOS}}]$, where $f_{\text{EOS}}$ is an indicator function for the final waypoint in the sequence. The final hidden state of the bi-directional gated recurrent unit (GRU) encoder serves as a feature vector that represents the waypoint sequence. Before it is passed to the decoder, the feature vector is densified by the VAE to prevent memorization behavior and force the model to summarize the waypoint sequence in a low-dimensional latent space [33]. Additionally, the hidden encoder states are used to generate content-based attention information that helps guide the decoder [19]. The feature vector and attention information are fed to the decoder network, which is a basic GRU [34].

The decoder output is passed through a fully-connected neural network and then converted to the latent time allocation $\tilde{\mathbf{x}}_t$, the latent smoothness weights $\tilde{\mathbf{x}}_w$, and the reconstructed waypoints $\mathbf{x}_p^{\text{out}}$

through additional separate fully-connected networks. Finally, a softmax (i.e., normalized exponential) function is used to make sure that the output time allocation $\mathbf{x}_t$ and the smoothness weights $\mathbf{x}_w$ are positive. Motivated by examples where performance on a primary task is improved by training on additional auxiliary tasks [35, 36, 37], we also train the model to minimize the reconstruction loss $\mathcal{L}_{\text{Recon}} = \left\| \mathbf{x}_p^{\text{in}} - \mathbf{x}_p^{\text{out}} \right\|^2$ of the waypoints. The similarity in feature space between the waypoints and the time allocation and smoothness weights enables the reconstruction loss minimization to guide the training on the primary tasks and to articulate the latent feature space. Indeed, we observed that adding this auxiliary task improves the stability of the training process.

### 3.3 Pretraining: Supervised Learning with Snap Minimization

In the first learning phase, the seq2seq model is pretrained using (4). While the resulting time allocations are not time-optimal, they provide a decent starting point for the learning process. Before generating the time allocations, we perform a grid search to find the penalty weights $\mu_r$ and $\mu_\psi$ that result in the fastest feasible trajectories on a subset of the waypoint sequences dataset. In order to further reduce trajectory times, we considered adding a weighted regularization term [5, 38]. However, through a grid search using a data subset, we found that there was no constant weight that improves the average performance, i.e., that did not also lead to many slower trajectories.

Based on the minimum-snap time allocations $\mathbf{x}_t^{\text{MS}}$, the seq2seq model is trained by minimizing

$$\mathcal{L} = \mathcal{L}_{\text{Recon}} + \mathcal{L}_{\text{ELBO}} + \left\| \mathbf{x}_t - \mathbf{x}_t^{\text{MS}} \right\|^2 + \left\| \mathbf{x}_w - \mathbf{x}_w^{\text{MS}} \right\|^2, \tag{8}$$

where $\mathcal{L}_{\text{ELBO}}$ is the evidence lower bound (ELBO) loss from the VAE, and $\mathbf{x}_t$ and $\mathbf{x}_w$ are the softmax-normalized outputs of the seq2seq model. In effect, (4) uses uniform smoothness weights, but training the model with constant $\mathbf{x}_w^{\text{MS}}$ converges to zero neural network weights because the softmax function normalizes the zero output to a uniform $\mathbf{x}_w$. In order to avoid this behavior—which causes issues in the next learning phase because all gradients will be zero—we set $\mathbf{x}_w^{\text{MS}}$ to the inverse of the time allocation, i.e., $x_{w,i}^{\text{MS}} = m(\sum_{j=1}^m x_{t,i}^{\text{MS}}/x_{t,j}^{\text{MS}})^{-1}$. Intuitively, these weights prioritize snap minimization on segments with small time allocation, which makes sense because these segments are flown faster on average and often require relatively increased smoothness to remain feasible.

### 3.4 Guiding: Supervised Learning with Bayesian Optimization

In the second training phase, we improve the model by using time-optimal time allocations and smoothness weights obtained for a subset of the dataset by solving (6) with BO. The BO algorithm is initialized using the solution from (4) and efficiently samples the $2m$-dimensional space of time allocations and smoothness weights to build a surrogate model of the feasibility constraint $\mathcal{P}_T$ [9]. We further improve efficiency by combining analytical and numerical feasibility evaluations using multi-fidelity BO, as described in Section 4.2. Based on the optimal time allocations $\mathbf{x}_t^{\text{BO}}$ and smoothness weights $\mathbf{x}_w^{\text{BO}}$ obtained using MFBO for a portion of waypoint sequences dataset, the seq2seq model is trained by minimizing the following loss function:

$$\mathcal{L} = \mathcal{L}_{\text{Recon}} + \mathcal{L}_{\text{ELBO}} + \left\| \mathbf{x}_t - \mathbf{x}_t^{\text{BO}} \right\|^2 + \left\| \mathbf{x}_w - \mathbf{x}_w^{\text{BO}} \right\|^2, \tag{9}$$

which has similar form as (8).

### 3.5 Generalizing: Reinforcement Learning

In the third and final phase, we use reinforcement learning to generalize the seq2seq model that was overfitted to the BO data subset. A Markov decision process is formulated with the preceding waypoint $x_{p,i-1}^{\text{out}}$ as the current state $s_i$ and the latent time allocation $\tilde{x}_{t,i}$ and latent smoothness weight $\tilde{x}_{w,i}$ as the action $a_i$. At the end of each episode, consisting of a variable-length sequence of $m+1$ waypoints, the minimum feasible trajectory duration $T_{\mathbf{x}_t, \mathbf{x}_w}$ for the normalized time allocation $\mathbf{x}_t$ and smoothness weights $\mathbf{x}_w$ is computed. The episode reward is the improvement of the trajectory time based on $\mathbf{x}_t$ and $\mathbf{x}_w$, compared to the baseline $T_{\text{MS}}$:

$$r(\mathbf{x}_t, \mathbf{x}_w) = 1 - T_{\mathbf{x}_t, \mathbf{x}_w}/T_{\text{MS}}. \tag{10}$$

Since the reward is only obtained a the end of the episode, we estimate the discounted reward at the $i$-th step, as $r_i = \sum_{j=i}^m \gamma^{j-i} r(s_j, a_j) = \gamma^{m-i} r(\mathbf{x}_t, \mathbf{x}_w)$, where $\gamma \in [0,1]$ is the discount factor.

We use the REINFORCE algorithm [39] to find the policy $\pi_\theta^*$ that maximizes the expected reward, i.e., $\pi_\theta^* = \arg\max_\theta \mathbf{E}_{\tilde{a}_1,\cdots,\tilde{a}_m \sim \pi_\theta(a_1,\cdots,a_m)} [r_m(\tilde{a}_1, \cdots, \tilde{a}_m)]$. During exploration, the action $\tilde{a}_i \sim \mathcal{N}(a_i, \sigma_{rf})$ is sampled from the normal distribution around the model output $a_i$. The objective function $\mathcal{L}_\theta = \sum_{i=1}^m r_i \log \pi_\theta(\tilde{a}_i|a_i)$ is estimated with batches of $N_{\text{batch}}$ sequences, as follows:

$$\mathcal{L}_\theta = N_{\text{batch}}^{-1} \sum_{k=1}^{N_{\text{batch}}} \sum_{i=1}^m (r_t - r_b) \log \pi_\theta(\tilde{a}_i|a_i), \tag{11}$$

where the baseline reward $r_b = N_{\text{batch}}^{-1} \sum_{k=1}^{N_{\text{batch}}} \sum_{i=1}^{m_k} r_i$ is subtracted to reduce reward variance. Moreover, all rewards are clipped to $[-0.5, 0.5]$ for further variance reduction. The objective function (11) is minimized together with the reconstruction loss and the VAE variational loss, i.e.,

$$\mathcal{L} = \mathcal{L}_{\text{Recon}} + \mathcal{L}_{\text{ELBO}} + \mathcal{L}_\theta. \tag{12}$$

## 4 Experimental Results

In order to validate the performance of our proposed algorithm, we trained the seq2seq model, evaluated the trajectories generated by the model, and tested these trajectories in flight experiments. Specifically, we trained two types of models: (i) using analytical feasibility evaluations based on the reference motor speeds obtained from the simplified quadrotor dynamics differential flatness transform, and (ii) using numerical evaluations of trajectory-tracking accuracy in a six-degree-of-freedom (6DOF) flight dynamics simulation. All training data consists of sequences of five to fourteen waypoints that are scaled to be used in a 10 m × 10 m × 10 m space, i.e., $L_{\text{space}} = 10$ m. Additional implementation details and experimental results can be found in the appendix and at https://youtu.be/1CD6UH8G7AY.

### 4.1 Differential Flatness Feasibility Constraint

We define the feasible set $\mathcal{P}_T$ as all trajectories that can be flown without violating the admissible motor speeds [0, 2200] rad/s according to idealized quadrotor dynamics differential flatness transform from [4]. Based on this set, we train two models: one that considers only waypoint positions and not yaw (i.e., constant yaw at zero), and one for a tangential (i.e., forward-facing) yaw reference (see Section 3.1). The pretraining and generalization stages were each terminated after 150 training epochs based on convergence of the training loss and the intermediate guiding stage was terminated after 1000 epochs when the performance gap between the training and validation data reached a set threshold. The BO stage requires more training epochs because it uses a much smaller training dataset than the other stages. Figure 1b shows the improvement over the three training stages for the model with forward-facing yaw reference. It can be seen that during the first stage the model approaches the performance of the minimum-snap labels. The BO data subset used in the second learning stage has superior labels, due to the more sophisticated optimization of (6). Overfitting to this dataset initially results in decreased performance on the validation dataset, but improves efficiency by guiding the RL algorithm. In the final stage, the model is generalized to consistent performance on the validation data. Moreover, Table 1 shows that all fully-trained models actually outperform the BO labels that were obtained using a limited number of iterations to curb computational cost.

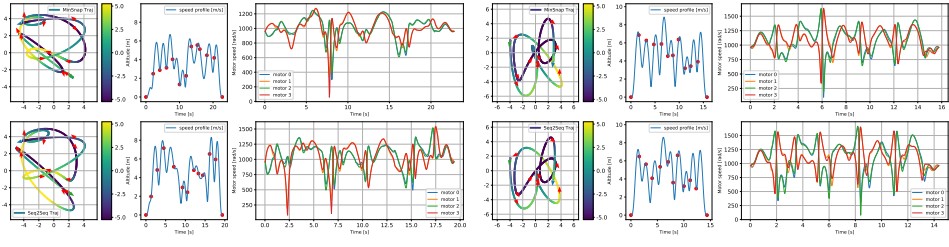

(a) 95th percentile (14.94 % improvement)       (b) 75th percentile (8.88 % improvement)

Figure 2: Trajectories obtained using minimum-snap optimization (top row) and the proposed seq2seq model (bottom row) with reference motor speeds.

With an average inference time of 0.01 s, our model is able to generate trajectories much faster than minimum-snap optimization algorithms, which require at least several seconds to solve the

Table 1: Trajectory time reduction relative to minimum-snap trajectories for BO labels obtained using BayesOpt on (6), and for the seq2seq model after Phase 2 and Phase 3 of learning. *BO set* refers to the data subset with BO labels; *Val. set* refers to the validation dataset.

| Feasibility constraint | | Differential flatness | | | | 6DOF Simulation |
|---|---|---|---|---|---|---|
| Yaw reference | | Constant | | Forward-facing | | Forward-facing |
| BO labels | | 5.375 % | | 4.746 % | | 5.552 % |
| Phase 2 (BO) | BO set \| Val. set | 3.884 % | -2.682 % | 4.178 % | -2.368 % | 5.048 % \| -2.582 % |
| Phase 3 (RL) | BO set \| **Val. set** | 6.971 % | **6.638 %** | 6.191 % | **6.031 %** | 7.169 % \| **7.291 %** |

Table 2: Trajectory time reduction by the seq2seq (s2s) model relative to minimum-snap (MS) trajectories for the validation data set with forward-facing yaw and using the differential flatness feasibility constraints. The bottom four rows contain data for the sample (i.e., the waypoint sequence and corresponding trajectory) at the percentile rank.

| Percentile | | 95th | 75th | 7th | 1st |
|---|---|---|---|---|---|
| Time reduction | | 14.94 % | 8.88 % | 1.08 % | −3.79 % |
| Waypoints | | 13 | 12 | 11 | 7 |
| Duration [s] | MS | 22.8 | 15.8 | 15.9 | 9.1 |
| Top speed [m/s] | MS \| s2s | 6.7 \| 8.3 | 8.8 \| 8.6 | 6.9 \| 7.7 | 9.1 \| 8.4 |
| Length [m] | MS \| s2s | 83.4 \| 87.8 | 71.9 \| 67.1 | 62.1 \| 67.7 | 38.5 \| 39.5 |
| Smoothness cost | (s2s / MS) | 1.6 | 0.8 | 1.2 | 1.2 |

nonconvex optimization (4). As shown in Table 2, the trained model also outperforms minimum-snap in terms of trajectory time on 93 % of the validation dataset. The improvement is greatest for trajectories with a large number of waypoints, where the seq2seq model has most options to modify the trajectory. The bottom row of Table 2 shows that the seq2seq trajectories have a higher smoothness cost than those from the minimum-snap method, i.e., they are less smooth and more aggressive. In practice, the proposed model generates lengthier paths that can be flown at a higher speed or lowers the flight speed to allow tighter turns, as shown in respectively Fig. 2a and Fig. 2b.

## 4.2 6DOF Simulation Feasibility Constraint

We now consider the feasible set $\mathcal{P}_T$ that contains all trajectories that can be flown with less than 20 cm position tracking error and 15 deg yaw tracking error based on a 6DOF flight dynamics and inertial measurement [40] and flight control [41] simulation. During the pretraining and RL stages, simulation is used for all feasibility evaluations, whereas it is combined with low-fidelity differential flatness evaluations during MFBO label generation. Training in simulation enables our model to consider more realistic feasibility constraints that incorporate vehicle aerodynamics, sensor noise, control system limitations etc. It also permits increasingly aggressive trajectories, including motor saturation, provided that an acceptable tracking error can be maintained. Consequently, our seq2seq model further exploits the vehicle capabilities, leading to faster trajectories. While the first two learning stages are again terminated after respectively 150 and 1000 epochs, we only perform three RL epochs in the final stage due to the increased computational cost of improvement evaluation. Despite this reduced RL training phase, our model for forward-facing yaw trajectories still outperforms minimum-snap trajectory generation on 87 % of the dataset with an overall average improvement of 7.3 %, as shown in Table 1.

We randomly select ten waypoint sequences with corresponding minimum-snap and seq2seq trajectories. Over a series of flight tests, we uniformly scale the corresponding time allocations until the aforementioned tracking error constraints are active. As shown in Table 3, the seq2seq time allocation ratios also result in quicker real-world trajectories. Figure 3a compares an example seq2seq trajectory with the corresponding minimum-snap trajectory. While the trajectory from our model has a lower top speed, it has tighter, more aggressive turns that utilize more of the motor speed capacity, leading to a 22 % reduction in trajectory duration.

In addition, we choose two sequences that contain several waypoints linked to moving obstacles, as shown in Figure 3b. The proposed Seq2Seq model is evaluated by updating the trajectory in real-time as the locations of these waypoints change. When compared to a real-time baseline, which reuses the initial trajectory's minimum-snap time allocation, the proposed model produces more

compact trajectories with shorter flight times. Detailed experimental results are available in the accompanying video.

Table 3: Length and duration of minimum-snap (MS) trajectories with simulation (sim) and real-world (exp) feasibility constraints, as well as duration improvements by seq2seq (s2s) model.

| Trajectory | | | 1 | 2 | 3 | 4 | 5 | 6 | 7 | 8 | 9 | 10 |
|---|---|---|---|---|---|---|---|---|---|---|---|---|
| Length [m] | MS | sim | 26.5 | 30.5 | 30.3 | 44.0 | 47.3 | 41.3 | 52.7 | 66.1 | 50.1 | 70.4 |
| Duration [s] | MS | sim | 7.2 | 9.7 | 9.3 | 13.2 | 13.0 | 12.9 | 18.4 | 18.9 | 16.4 | 21.5 |
| Imp. [%] | s2s | sim | 11.9 | 5.6 | 13.7 | 9.4 | 13.5 | 11.3 | 8.1 | 12.0 | 8.6 | 18.7 |
| Duration [s] | MS | exp | 5.8 | 7.7 | 8.4 | 10.6 | 12.7 | 11.7 | 13.6 | 15.8 | 16.2 | 20.2 |
| Imp. [%] | s2s | exp | 4.3 | 1.3 | 10.5 | 10.4 | 4.2 | 14.0 | 9.2 | 5.2 | 8.9 | 22.1 |

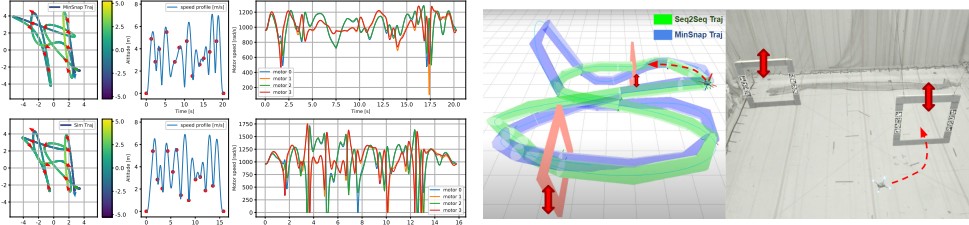

(a) Traj. 10 (22.11 % real-world improvement).  (b) Online trajectory adaptation.

Figure 3: Trajectories obtained using minimum-snap optimization (left, top row) and the proposed seq2seq model (left, bottom row) with reference motor speeds, and real-time trajectory generation through moving obstacles (right).

# 5 Conclusion

## 5.1 Contributions

We presented a seq2seq model that generates time-optimal quadrotor trajectories from waypoint sequences. In order to train the model, we introduced a novel three-stage learning process that combines the advantages of supervised learning, Bayesian optimization, and reinforcement learning. The resulting model outperforms minimum-snap trajectory generation in both computation time and time-optimality of the generated trajectories in simulation and real-world flight experiments.

## 5.2 Limitations

The main limitation of the proposed method is its lack of theoretical performance guarantees. While the trained model outperforms minimum-snap trajectory generation for the vast majority of the validation dataset, it generates slower trajectories for a small subset of waypoint sequences, e.g., the edge case where all but one waypoints are located very closely together. We expect that utilizing ensemble modeling, i.e., training several models with varying parameters, or increasing the size of the data subset labeled with BO may relieve this issue.

Additionally, the learning process is computationally expensive. Using a multi-core implementation on a powerful desktop computer, the complete process of data labeling and training described in Section 4 took on the order of several weeks for the differential flatness feasibility constraints and on the order of several months for the 6DOF simulation. In particular, the costly computation of RL rewards with the 6DOF simulation may be sped up by using a multi-fidelity formulation that incorporates cheap low-fidelity evaluations from the differential flatness feasibility model, similar our approach for BO.

Finally, our model still needs some modifications to make it suitable for online motion planning. While it can generate trajectories in real time, it is currently limited to waypoint sequences that start and end in static hover. Our future work includes extending the seq2seq model to allow dynamic start and end points with nonzero velocity, yaw rate, acceleration etc., and deploying the model on an embedded system for online motion planning.

**Acknowledgments**

This work was partly supported by the Army Research Office through grant W911NF1910322.

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
