# OpenReview forum: "Real-Time Generation of Time-Optimal Quadrotor Trajectories with Semi-Supervised Seq2Seq Learning"
_robot-learning.org/CoRL/2022/Conference — CoRL 2022 Poster_

### Official Review · Reviewer_a94g · 2022-07-30

**Originality:** Good
**Technical Quality:** Very Good
**Clarity Of Presentation:** Excellent
**Impact:** 3

**Recommendation:**

Weak Accept: I recommend accepting the paper, but will not argue for my recommendation if the majority of other reviewers have a different opinion.

**Summary:**

The trajectory optimization problem for a quadrotor to fly through multiple way-points in as short a time as possible has very complex dynamics. The problem is how to deal with the varying number of way-points and large computational cost, especially when the trajectory must be generated in real-time. Usually, simplified dynamics models or polynomial approximations of limited intervals are used, but these methods have limited performance. In this paper, the authors have succeeded in constructing a seq2seq model that can generate trajectories with a lower computational cost by approximating the exact solution with a DNN model and then generalizing the trained DNN model by reinforcement learning.

**Issues:**

- In order to discuss the scalability of the proposed method, specifics on the network size of the seq2seq model and the details of the GPGPU environment used in the experiments are required.
- In addition to real-time performance, it would be better to discuss battery consumption.


**Quality Of The Limitations Section:**

Limitations are addressed clearly

**Reviewer Expertise:**

4: The reviewer is confident but not absolutely certain that the evaluation is correct

**Robotics Focus:**

Sufficient demonstration on hardware

**Strengths And Weaknesses:**

- The paper is consistent and clearly presented from the problem statement to the proposal of a solution and its evaluations through experiments on real systems.
- The most significant contribution of the paper is the proposal of a framework for solving the problem stated in the abstract, but the reader does not have sufficient information necessary to implement the proposed framework. For example, the network structure of the seq2seq model and the method of preparing training data.


**Summary Of Recommendation:**

The consistency of the paper, the clarity of the presentation, and the implementation in real environments are highly evaluated, but on the other hand, the novelty of the learning algorithm is not high, and the evaluation on generalization performance is not sufficient. Therefore, it was evaluated one rank lower than the highest.

---

> ### Author Response · Authors · 2022-08-23
> **Thanks a lot for reviewing our paper**
>
> Thanks a lot for reviewing our paper.
>
> Due to the page limit, we attached the details of our seq2seq model architecture in the appendix. It includes the exact size of DNN layers, training parameters, and the size of the dataset used in each training procedure. An Nvidia Titan V GPU was used for training the model, but the GPU was not the main bottleneck since our DNN model consists of only 1~2 layers with a 256 hidden size. The main bottleneck of training comes from the RAM size since we have to parallelize the high-fidelity simulation as much as we can. We also included details on the training data and the full experimental results in the appendix.
>
> Our method focuses on generating time-optimal trajectories, i.e., racing trajectories where the objective is to be as fast as possible and the battery usage required to fly the trajectory is less relevant. Additionally, the battery consumption due to onboard computation is negligible compared to the battery consumption by the four motors. (Maximum power consumption by a Jetson TX2 is roughly 10W, while a 752 gram quadcopter requires 130 W just to hover. [1])
>
> [1] Bauersfeld, Leonard, and Davide Scaramuzza. "Range, Endurance, and Optimal Speed Estimates for Multicopters." IEEE Robotics and Automation Letters 7.2 (2022): 2953-2960.

---

### Official Review · Reviewer_7heK · 2022-08-01

**Originality:** Fair
**Technical Quality:** Good
**Clarity Of Presentation:** Fair
**Impact:** 3

**Recommendation:**

Weak Reject: I recommend rejecting the paper, but will not argue for my recommendation if the majority of other reviewers have a different opinion.

**Summary:**

The work presents an approach for real time inference of near-optimal time allocation for trajectory segments of minimum snap trajectories for quadrotors. To this end, the work utilizes a seq2seq model that is trained in three stages i) imitation learning from suboptimal trajectories ii) Bayesian Optimization using a smaller portion of optimal trajectories iii) RL refinement. To demonstrate the quality of the proposed method, the trajectories generated with this method are tested in simulation and on a real quadrotor platform.

**Issues:**

See Strengths & Weaknesses

**Quality Of The Limitations Section:**

Limitations are addressed clearly

**Reviewer Expertise:**

2: The reviewer is willing to defend the evaluation, but it is quite likely that the reviewer did not understand central parts of the paper

**Robotics Focus:**

Sufficient demonstration on hardware

**Strengths And Weaknesses:**

Praise:
* The work follows a generally interesting avenue of real time trajectory generation for very computation constrained platforms, such as drones.

* The work demonstrates results using a real quadrotor platform.


Concerns:
   * To be honest I am not sure what specifically is called a “trajectory” in the context of this work, i.e. what parameters of the trajectory are minimized, what is trajectory representation? It does seem that the output of the seq2seq model is simply time allocations and smoothness weights, but how is trajectory itself generated, i.e. polynomial parameters or maybe quadrotor thrusts/velocity commands?

   * What are the assumptions in this work? From what I understand the work assumes not only knowledge of the full dynamics model of the quadrotor, but also presence of a tracking controller that can go between provided trajectory key points, given the budget of time. The combination of both seems to constitute the feasibility set P_T.

   * The method is rather complex using a 3 stage training procedure and 3 different approaches, which is definitely error-prone and, likely, introduces lots of tuning and hyperparameters. Modern RL methods under similar assumptions can solve for direct quadrotor commands (with implicit planning) and do it in hours (worst case days), not 4 weeks as stated in the Limitation section while showing very compelling results for various tasks in quadrotor flight (see [5] below just as an example).

   * On the other hand,  work [1] presents seemingly strong baseline for simultaneous planning and tracking (there is generally a larger body of work from the same lab for time-optimal, high-speed robust quadrotor flight). It surely uses simplified assumptions on the planning part, yet demonstrates high performance even beating human pilots. The work under consideration does not seem to have an advantage here since it also lacks specific guarantees (it uses NN trained through BC and RL) and, I believe, relies on the accuracy of the model. Hence, it would be interesting to have some comparison or strong justification why such comparison is not appropriate.

   * It is not completely clear why the authors need smoothness weights and time allocation at the same time. Both represent just normalized (over the entire trajectory) scaling of the cost per segment, i.e. two scaling parameters seem redundant.

   * Literature overview is not complete, please, include (at least) these works with explanation of differences/advantages of your work:
      * [1] Angel et al. Model Predictive Contouring Control for Time-Optimal Quadrotor Flight.
      * [2] Panicka et al. Learning Minimum-Time Flight in Cluttered Environments.
      * [3] Romero et al. Time-Optimal Online Replanning for Agile Quadrotor Flight.
      * [4] Loquercio et al. Learning High-Speed Flight in the Wild.
      * [5] Molchanov et al. Sim-to-(Multi)-Real: Transfer of Low-Level Robust Control Policies to Multiple Quadrotors.

**Summary Of Recommendation:**

As stated in previous sections, I have problems with understanding the significance of the potential impact this work can make, especially considering missed references in the related work section.
I am, clearly, not an expert in the area of planning for minimum snap quadrotor trajectories, hence, maybe, after some strong arguments I might change my opinion. But, for now, I am inclined toward rejection of the paper.

---

> ### Author Response · Authors · 2022-08-23
> **Thanks a lot for reviewing our paper (1)**
>
> Thanks a lot for reviewing our paper. We will address the raised concerns by providing additional clarifications of our problem formulation and approach.
>
> ### Trajectory representation
> As correctly stated by the reviewer, the output of the seq2seq model are the time allocation and smoothness weights. The convex program (2) is used to obtain the corresponding trajectory, which is represented by four piecewise polynomial functions that give the x, y, and z position, and the yaw as a function of time.
>
> Our seq2seq model can be understood as an advanced optimal speed profile adaptation. It gives the optimal time allocation over the trajectory segments between waypoints. It also somewhat changes the shape of the trajectory, in particular through the smoothness weights, to determine the aggressiveness of direction changes in each segment.
>
> ### Feasibility set P_T
> For the simple analytical model described in Section 4.1, we define the feasible set P_T as all trajectories that can be flown without violating the admissible motor speeds according to the idealized quadrotor dynamics differential flatness transform. In the 6DOF simulation (described in Section 4.2) that is used to generate the trajectories for the flight experiments, a trajectory is feasible if it can be flown within a specified tracking error bound. This more holistic approach to feasibility indeed also considers the control algorithm as well as imperfect IMU measurements, actuation bandwidth etc., and is evaluated using numerical simulations.
>
> The definition of the feasibility set distinguishes our method from many existing formulations. We avoid using simplified predefined constraints and purely determine the feasibility boundary from the evaluation data. This is especially relevant when considering very aggressive (drone racing) trajectories, where just linear velocity or acceleration bounds are not realistic constraints. For instance, even with saturated motor speeds, it may be possible to accurately track a trajectory; and, on the other hand, imperfect estimation and actuation may lead to large tracking errors despite feasible reference motor speeds.
>
> ### DNN training time
> We understand that the three-phase training procedure may look complicated, but in practice, we found that it actually increases the robustness and efficiency of the learning process. The first and second phases are pre-training the model. Therefore, their hyperparameter tuning has a limited effect on the final performance of the fully trained model. The third phase (RL training) is guided to high-quality solutions by the pre-training, making it less error-prone and decreasing sensitivity to hyperparameter tuning. Additionally, the pre-training strongly increases the efficiency of the RL phase.
>
> Compared to other RL works, our algorithm takes longer computation mainly due to the complexity of the simulation, which accounts for more than 95% of the total training time. In order to simulate flight dynamics with a vehicle aerodynamics model, but also the effects of imperfect flight control, IMU measurement noise, and actuation bandwidth, we run a relatively complex simulation with a small timestep. Ultimately, this enables our model to avoid conservative simplifications and allows the generation of trajectories that fully exploit the vehicle's capability.

---

> > ### Author Response · Authors · 2022-08-23
> > **Thanks a lot for reviewing our paper (2)**
> >
> >
> > ### Literature review and comparison
> > Thanks a lot for adding great references. We will include these in our literature review. We would like to clarify the problem we address and how it is complementary to the cited works.
> >
> > Our work considers the problem of online time-optimal trajectory planning, specifically, real-time generation of the fastest dynamically feasible reference trajectory that attains any given sequence of waypoints.
> >
> > [1] presents a combined planning and control strategy that does however still require a reference path (e.g., like the one provided by our algorithm).
> >
> > [2] presents a NN online planning algorithm that can replan trajectories between waypoints while avoiding obstacles. However, the training of the model is dependent on the waypoints and the environment. It does not generalize to unseen waypoint sequences. On the other hand, our trained model can generate a trajectory for any sequence of waypoints through a single inference.
> >
> > [3] presents a sampling-based method that generates a velocity search graph between two waypoints (or gates) and finds the optimal velocity profile with the Dijkstra search. However, the velocity search graph is generated without accurate modeling of the vehicle dynamics, leading to infeasible or conservative trajectories.
> >
> > [4] presents a learning-based planner which is trained with imitation learning. The main focus of this paper is generating a collision-free sequence of waypoints based on depth estimation. Our work focuses on obtaining the time-optimal trajectory given such a sequence of waypoints.
> >
> > [5] presents a low-level controller that takes a state estimate and a reference (e.g., as provided by our trajectory generation algorithm) and outputs the closed-loop quadcopter control inputs.
> >
> > ### Smoothness weights
> > Both the smoothness weights and time allocation are used in the quadratic program (5) and affect the obtained trajectory in different ways. Hence, adding the smoothness weights gives the model additional freedom to optimize the trajectory. For instance, for a given time allocation, subsequent waypoints can be connected with a fast and long trajectory segment, or with a shorter segment flown at decreased speed. Typically, a high smoothness cost would correspond to the former option, while a lower cost is incurred for the latter option.

---

### Official Review · Reviewer_SHLP · 2022-08-01

**Originality:** Good
**Technical Quality:** Good
**Clarity Of Presentation:** Very Good
**Impact:** 4

**Recommendation:**

Weak Accept: I recommend accepting the paper, but will not argue for my recommendation if the majority of other reviewers have a different opinion.

**Summary:**

1. This paper proposes a data-driven method for time-optimal trajectory generation for quadrotors.
2. Supervised pretraining and bayesian optizatimization & reinforcement learning are combined, which is interesting. The three stage training scheme is one of the core contribution.
3. Superior perfromance is validated both by simulation and onboard tests.

**Issues:**

The four weak points in the previous section may all be justified in the response.

**Quality Of The Limitations Section:**

Additional details required

**Reviewer Expertise:**

5: The reviewer is absolutely certain that the evaluation is correct and very familiar with the relevant literature

**Robotics Focus:**

Sufficient demonstration on hardware

**Strengths And Weaknesses:**

### Strengths
1. Three-stage training scheme is interesting.
2. Modeling min-time trajectory generation problem as a seq2seq learning problem should be a good contribution to the community.

### Weaknesses
1. **The literature review and experimental comparisons are not sufficient**. The authors may consider comparing your methods with the following: 1) "Optimal Time Allocation for Quadrotor Trajectory Generation" by Fei Gao et al; 2) Time-Optimal Planning for Quadrotor Waypoint Flight by David Scaramuzza et al; They are both open-sourced and highly related to your topic. Minimum snap (MS) is now a relatively weak baseline and 20% faster than MS may not be a very supersing achievement.

2. **Why using BO for the labeling of the sceond stage?**. BO may be too slow for this task and may limit the scale of the dataset for the second stage. The authors may consider using stronger methods as listed in 1. It can also be expected that the performance of the proposed pipeline is affected by the scale of the dataset for the second stage. A stronger method for the second stage may lead to a much stronger performance.

3. **Still some gap to online motion planning** The authors do not incorporate the proposed method in a planning system in the wild. It will be much challenging for the proposed method to work generally in the wide. It is still unknown whether it can generalize that well. Although this point is also listed in the Limitations section, the reviewer thinks this is still a marjor weak point.

4. **The scalability problem**. The authors mention that, even for this model in limited ODD, it still takes weeks to train. Many intermediate methods are time consuming. The problem comes that whether the pipeline is still feasible for a motion planning method in the wild (the scale of the dataset should be much much larger than the current one). The authors may provide further thoughts on this issue.


**Summary Of Recommendation:**

The idea is interesting, but still some gap to a solid contribution.

---

> ### Author Response · Authors · 2022-08-23
> **Thanks a lot for reviewing our paper**
>
> Thank you for the insightful review! We have addressed the four perceived weaknesses below and will revise our submission with the following comparisons and clarifications.
>
>
> ### Literature review and experimental comparisons
> Our model is designed to output the near-optimal time allocation and weights for trajectory generation s.t. (2), similar to the min-snap baseline (4). Hence, our experimental comparison focused on showing that our algorithm produces (at a fraction of the computational cost) high-quality time allocations that are actually faster (i.e. more time-optimal) than the min-snap baseline algorithm. Given this perspective, we consider min-snap the most relevant baseline for our learned model.
>
> Nonetheless, we agree that both mentioned papers are highly related to the topic of our work and will include both in the literature review. Fei Gao et al. separate spatial and temporal optimization to obtain a convex optimization problem subject to simple linear velocity and acceleration constraints. Our model learns a more realistic feasibility constraint that incorporates vehicle dynamics. Therefore, our method may lead to less conservative trajectories. Scaramuzza et al. employ a shooting method to find optimal trajectories without topological constraints. The resulting algorithm is very computationally expensive, i.e., several hours per trajectory compared to a few milliseconds of inference time in our method, making it unsuitable for real-time application.
>
> ### BO labeling for the second learning phase
> We purposely selected BO for the second learning phase, because it efficiently models the dynamic feasibility constraints based on only evaluation data without assuming simplified constraints or models. This enables us to use a sophisticated simulation that avoids discrepancies in the idealized dynamics models that are used in the works mentioned above, e.g., in practice, saturated motor speeds do not always lead to large trajectory tracking error.
>
> Additionally, we emphasize that the BO phase only serves to guide the RL phase. Hence, we focus on a relatively small data set with high-quality solutions rather than a generally representative large data set. We found that the scale of the BO dataset (0.25% of the training dataset) is sufficient to effectively guide the RL phase towards general high-quality solutions, so we do not anticipate increasing the size of the second-phase data set will significantly improve overall performance.
>
> ### Online motion planning
> In our flight experiments, we see good performance on the real vehicle for a large variety of randomly generated waypoint sequences. The trajectory generation algorithm is independent of the source of the waypoints, so we are confident that it will attain similar performance for waypoint sequences that are generated online (e.g., using output from task planning and mapping algorithms).
>
> As described in the limitations section, there are two main steps towards online implementation: deployment on an embedded system, and extension to allow dynamic start and end points. The deployment is relatively straightforward since our algorithm is specifically designed for high-rate online planning with limited resources and only the inference needs to be implemented on the embedded platform.
>
> The nonzero derivative constraints at the start and end points can be included in the seq2seq model as a form of a context vector. We expect that this can be achieved by replacing the autoencoder structure used in this paper with the conditional generative model [1]. We also notice that this online planning algorithm can be considered as a receding horizon planner. This implies that we can reduce the maximum number of waypoints to limit the dimensionality of the data and thereby the computational cost.
>
> ### Scalability
> The main reason for the long computation time is the complexity of the simulation. To simulate the effects of the flight control algorithm, IMU measurement noise, actuation bandwidth, and vehicle aerodynamics, we run a relatively complex simulation with a small timestep, accounting for over 95% of the total training time. We use a DNN model with 1~2 fully-connected layers, so (with GPU parallelization) its update time is almost negligible. Moreover, the actual number of DNN updates is smaller than in typical RL examples, since we improve the training efficiency through the three-phase learning procedure. We expect that improving the simulation efficiency and perhaps trading some accuracy for computational efficiency can reduce the overall computation time by an order of magnitude, greatly improving scalability.
>
> [1] Sohn, Kihyuk, Honglak Lee, and Xinchen Yan. "Learning structured output representation using deep conditional generative models." Advances in neural information processing systems 28 (2015)

---

> > ### Comment · Reviewer_SHLP · 2022-08-27
> > **Response to Authors**
> >
> > Dear Authors,
> >
> > Thanks for your efforts and detailed explaination. The four points are clarified. The reviewer would like to keep the rating unchanged.

---

### Meta-Review · Area_Chair_HtzD · 2022-08-14

**Recommendation:** Accept (Poster)
**Confidence:** 3

**Metareview:**

The paper presents a data-driven method for solving the planning/control problem of quadrotors with multiple way-points.

Strengths:
- Modeling min-time trajectory generation problem as a seq2seq learning problem should be a good contribution to the community (SHLP)
- Evaluation on real system (a94g)

Weaknesses:
- insufficient literature review (SHLP)
- scalability (SHLP,a94g)
- wrong statements about comparison to prior art (7heK)
- missing baseline/comparison to e.g. Angel et al (7heK)
- an improvement over a single baseline that is not state of the art anymore (SHLP)

This is just an excerpt from the reviews, and I invite the authors to address all points raised by the reviewers.

Post-rebuttal:
Two reviewers suggest weak accept and one reviewer suggests weak rejection.
The authors have addressed some concerns in the response.
I ask the authors to make sure the manuscript is adapted to clarify the concerns itself.
I would also ask the authors to add ablation studies showing the contribution of the components. What if parts of the 3 learning stages are omitted, for instance? This is a borderline paper, but given the reviewer scores, I am leaning towards acceptance.
I leave the final decision to the PC.

---

> ### Author Response · Authors · 2022-08-23
> **Thanks a lot for reviewing our paper**
>
> We thank the reviewers and the area chair for the time and effort spent reviewing the paper. The insightful comments are helping us improve the paper revision and are addressed below each review.